# On Reward-Free Reinforcement Learning with Linear Function Approximation[*]

**Ruosong Wang**[†]      **Simon S. Du**[‡]      **Lin F. Yang**[§]      **Ruslan Salakhutdinov**[†]

## Abstract

Reward-free reinforcement learning (RL) is a framework which is suitable for both the batch RL setting and the setting where there are many reward functions of interest. During the exploration phase, an agent collects samples without using a pre-specified reward function. After the exploration phase, a reward function is given, and the agent uses samples collected during the exploration phase to compute a near-optimal policy. Jin et al. [2020] showed that in the tabular setting, the agent only needs to collect polynomial number of samples (in terms of the number states, the number of actions, and the planning horizon) for reward-free RL. However, in practice, the number of states and actions can be large, and thus function approximation schemes are required for generalization. In this work, we give both positive and negative results for reward-free RL with linear function approximation. We give an algorithm for reward-free RL in the linear Markov decision process setting where both the transition and the reward admit linear representations. The sample complexity of our algorithm is polynomial in the feature dimension and the planning horizon, and is completely independent of the number of states and actions. We further give an exponential lower bound for reward-free RL in the setting where only the optimal $Q$-function admits a linear representation. Our results imply several interesting exponential separations on the sample complexity of reward-free RL.

## 1 Introduction

In reinforcement learning (RL), an agent repeatedly interacts with an unknown environment to maximize the cumulative reward. To achieve this goal, RL algorithms must be equipped with exploration mechanisms to effectively solve tasks with long horizons and sparse reward signals. Empirically, there is a host of successes by combining deep RL methods with different exploration strategies. However, the theoretical understanding of exploration in RL by far is rather limited.

In this work we study the reward-free RL setting which was formalized in the recent work by Jin et al. [2020]. There are two phases in the reward-free setting: the exploration phase and the planning phase. During the exploration phase, the agent collects trajectories from an unknown environment without any pre-specified reward function. Then, in the planning phase, a specific reward function is given to the agent, and the goal is to use samples collected during the exploration phase to output a near-optimal policy for the given reward function. From a practical point of view, this paradigm is particularly suitable for 1) the batch RL setting [Bertsekas and Tsitsiklis, 1996] where data collection and planning are explicitly separated and 2) the setting where there are multiple reward function

---

[*]Correspondence to: Ruosong Wang <ruosongw@andrew.cmu.edu>, Simon S. Du <ssdu@cs.washington.edu>, Lin F. Yang <linyang@ee.ucla.edu>, Ruslan Salakhutdinov <rsalakhu@cs.cmu.edu>.

[†]Carnegie Mellon University

[‡]University of Washington, Seattle

[§]University of California, Los Angeles

of interest, e.g., constrained RL [Achiam et al., 2017, Altman, 1999, Tessler et al., 2018]. From a theoretical point view, this setting separates the exploration problem and the planning problem which allows one to handle them in a theoretically principled way, in contrast to the standard RL setting where one needs to deal with both problems simultaneously.

Key in this framework is to collect a dataset with sufficiently good coverage over the state space during the exploration phase, so that one can apply a batch RL algorithm on the dataset [Chen and Jiang, 2019, Agarwal et al., 2019, Antos et al., 2008, Munos and Szepesvári, 2008] during the planning phase. For the reward-free RL setting, existing theoretical works only apply to the tabular RL setting. Jin et al. [2020] showed that in the tabular setting where the state space has bounded size, $\widetilde{O}(\text{poly}(|\mathcal{S}||\mathcal{A}|H)/\varepsilon^2)$ samples during the exploration phase is *necessary and sufficient* in order to output $\varepsilon$-optimal policies in the planning phase. Here, $|\mathcal{S}|$ is the number of states, $|\mathcal{A}|$ is the number of actions and $H$ is the planning horizon.

The sample complexity bound in [Jin et al., 2020], although being near-optimal in the tabular setting, can be unacceptably large in practice due to the polynomial dependency on the size of the state space. For environments with a large state space, function approximation schemes are needed for generalization. RL with linear function approximation is arguably the simplest yet most fundamental setting. Clearly, in order to understand more general function classes, e.g., deep neural networks, one must understand the class of linear functions first. In this paper, we study RL with linear function approximation in the reward-free setting, and our goal is to answer the following question:

*Is it possible to design provably efficient RL algorithms with linear function approximation in the reward-free setting?*

We obtain both a polynomial upper bound and a hardness result to the above question.

**Our Contributions.** Our first contribution is a provably efficient algorithm for reward-free RL under the linear MDP assumption [Yang and Wang, 2019, Jin et al., 2019], which, roughly speaking, requires both the transition operators and the reward functions to be linear functions of a $d$-dimensional feature extractor given to the agent. See Assumption 2.1 for the formal statement of the linear MDP assumption. Our algorithm, formally presented in Section 3, samples $\widetilde{O}\left(d^3 H^6/\varepsilon^2\right)$ trajectories during the exploration phase, and outputs $\varepsilon$-optimal policies for an arbitrary number of reward functions satisfying Assumption 2.1 during the planning phase with high probability. Here $d$ is the feature dimension, $H$ is the planning horizon and $\varepsilon$ is the required accuracy.

One may wonder whether is possible to further weaken the linear MDP assumption, since it requires the feature extractor to encode model information, and such feature extractor might be hard to construct in practice. Our second contribution is a hardness result for reward-free RL under the linear $Q^*$ assumption, which only requires the optimal value function to be a linear function of the given feature extractor and thus weaker than the linear MDP assumption. Our hardness result, formally presented in Section 4, shows that under the linear $Q^*$ assumption, any algorithm requires exponential number of samples during the exploration phase, so that the agent could output a near-optimal policy during the planning phase with high probability. The hardness result holds even when the MDP is deterministic.

Our results highlight the following conceptual insights.

- **Reward-free RL might require the feature to encode model information.** Under model-based assumption (linear MDP assumption), there exists a polynomial sample complexity upper bound for reward-free RL, while under value-based assumption (linear $Q^*$ assumption), there is an exponential sample complexity lower bound. Therefore, the linear $Q^*$ assumption is *strictly weaker* than the linear MDP assumption in the reward-free setting.

- **Reward-free RL could be exponentially harder than standard RL.** For deterministic systems, under the assumption that the optimal $Q$-function is linear, there exists a polynomial sample complexity upper bound [Wen and Van Roy, 2013] in the standard RL setting. However, our hardness result demonstrates that under the same assumption, any algorithm requires exponential number of samples in the reward-free setting.

- **Simulators could be exponentially more powerful.** In the setting where the agent has sampling access to a generative model (a.k.a. simulator) of the MDP, the agent can query the next state $s'$ sampled from the transition operator given any state-action pair as input. In

the supplementary material, we show that for deterministic systems, under the linear $Q^*$ assumption, there exists a polynomial sample complexity upper bound in the reward-free setting when the agent has sampling access to a generative model. Compared with the hardness result above, this upper bound demonstrates an exponential separation between the sample complexity of reward-free RL in the generative model and that in the standard RL model. To the best our knowledge, this is the first exponential separation between the standard RL model and the generative model for a natural question.

## 1.1 Related Work

Practitioners have proposed various exploration algorithms for RL without using explicit reward signals [Chentanez et al., 2005, Schmidhuber, 2010, Bellemare et al., 2016, Houthooft et al., 2016, Tang et al., 2017, Florensa et al., 2017, Pathak et al., 2017, Tang et al., 2017, Achiam et al., 2017, Hazan et al., 2018, Burda et al., 2018, Colas et al., 2018, Co-Reyes et al., 2018, Nair et al., 2018, Eysenbach et al., 2018, Pong et al., 2019]. Theoretically, for the tabular case, while the reward-free setting is first formalized in Jin et al. [2020], algorithms in earlier works also guarantee to collect a polynomial-size dataset with coverage guarantees [Brafman and Tennenholtz, 2002, Hazan et al., 2018, Du et al., 2019a, Misra et al., 2019].[5] Jin et al. [2020] gave a new algorithm which has $\widetilde{O}(|\mathcal{S}|^2 |\mathcal{A}| \operatorname{poly}(H)/\varepsilon^2)$ sample complexity. They also proved a lower bound showing that the dependency on $|\mathcal{S}|$, $|\mathcal{A}|$ and $\varepsilon$ of their algorithm is optimal up to logarithmic factors. There are also extensions to related settings, e.g., [Tarbouriech et al., 2020, Zhang et al., 2020]. One of questions asked in [Jin et al., 2020] is whether their result can be generalized to the function approximation setting.

This paper studies linear function approximation. Linear MDP is the setting where both the transition and the reward are linear functions of a given feature extractor. Recently, in the standard RL setting, many works [Yang and Wang, 2019, Jin et al., 2019, Cai et al., 2020, Zanette et al., 2019] have provided polynomial sample complexity guarantees for different algorithms in linear MDPs. Technically, our algorithm, which works in the reward-free setting, combines the algorithmic framework in [Jin et al., 2019] with a novel exploration-driven reward function (cf. Section 3). Linear $Q^*$ is another setting where only the optimal $Q$-function is assumed to be linear, which is weaker than the assumptions in the linear MDP setting. In the standard RL setting, it is an open problem whether one can use polynomial number of samples to find a near-optimal policy in the linear $Q^*$ setting [Du et al., 2020]. Existing upper bounds all require additional assumptions, such as (nearly) deterministic transition [Wen and Van Roy, 2013, Du et al., 2019b].

## 2 Preliminaries

Throughout this paper, for a given positive integer $N$, we use $[N]$ to denote the set $\{1, 2, \ldots, N\}$.

### 2.1 Episodic Reinforcement Learning

Let $\mathcal{M} = (\mathcal{S}, \mathcal{A}, P, r, H, \mu)$ be a *Markov decision process* (MDP) where $\mathcal{S}$ is the state space, $\mathcal{A}$ is the action space with bounded size, $P = \{P_h\}_{h=1}^H$ where $P_h : \mathcal{S} \times \mathcal{A} \to \Delta(\mathcal{S})$ is the transition operator in level $h$ which takes a state-action pair and returns a distribution over states, $r = \{r_h\}_{h=1}^H$ where $r_h : \mathcal{S} \times \mathcal{A} \to [0, 1]$ is the deterministic reward function[6] in level $h$, $H \in \mathbb{Z}_+$ is the planning horizon (episode length), and $\mu \in \Delta(\mathcal{S})$ is the initial state distribution.

When the initial distribution $\mu$ and the transition operators $P = \{P_h\}_{h=1}^H$ are all deterministic, we say $\mathcal{M}$ is a *deterministic system*. In this case, we may regard each transition operator $P_h : \mathcal{S} \times \mathcal{A} \to \mathcal{S}$ as a function that maps state-action pairs to a states. We note that deterministic systems are special cases of general MDPs.

A policy $\pi$ chooses an action $a \in \mathcal{A}$ based on the current state $s \in \mathcal{S}$ and the time step $h \in [H]$. Formally, $\pi = \{\pi_h\}_{h=1}^H$ where for each $h \in [H]$, $\pi_h : \mathcal{S} \to \mathcal{A}$ maps a given state to an action.

The policy $\pi$ induces a trajectory $s_1, a_1, r_1, s_2, a_2, r_2, \ldots, s_H, a_H, r_H$, where $s_1 \sim \mu$, $a_1 = \pi_1(s_1)$, $r_1 = r_1(s_1, a_1)$, $s_2 \sim P(s_1, a_1)$, $a_2 = \pi_2(s_2)$, $r_2 = r_2(s_2, a_2)$, etc.

An important concept in RL is the $Q$-function. For a specific set of reward functions $r = \{r_h\}_{h=1}^H$, given a policy $\pi$, a level $h \in [H]$ and a state-action pair $(s, a) \in \mathcal{S} \times \mathcal{A}$, the $Q$-function is defined as

$$Q_h^\pi(s, a, r) = \mathbb{E}\left[\sum_{h'=h}^H r_{h'}(s_{h'}, a_{h'}) \mid s_h = s, a_h = a, \pi\right].$$

Similarly, the value function of a given state $s \in \mathcal{S}$ is defined as

$$V_h^\pi(s, r) = \mathbb{E}\left[\sum_{h'=h}^H r_{h'}(s_{h'}, a_{h'}) \mid s_h = s, \pi\right].$$

For a specific set of reward functions $r = \{r_h\}_{h=1}^H$, We use $\pi_r^*$ to denote an optimal policy with respect to $r$, i.e., $\pi_r^*$ is a policy that maximizes

$$\mathbb{E}\left[\sum_{h=1}^H r_h(s_h, a_h) \mid \pi\right].$$

We also denote $Q_h^*(s, a, r) = Q_h^{\pi_r^*}(s, a, r)$ and $V_h^*(s, r) = V_h^{\pi_r^*}(s, r)$. We say a policy $\pi$ is $\varepsilon$-optimal with respect to $r$ if

$$\mathbb{E}\left[\sum_{h=1}^H r_h(s_h, a_h) \mid \pi\right] \geq \mathbb{E}\left[\sum_{h=1}^H r_h(s_h, a_h) \mid \pi_r^*\right] - \varepsilon.$$

Throughout the paper, when $r$ is clear from the context, we may omit $r$ from $Q_h^\pi(s, a, r)$, $V_h^\pi(s, r)$, $Q_h^*(s, a, r)$, $V_h^*(s, r)$ and $\pi_r^*$.

## 2.2 Linear Function Approximation

When applying linear function approximation schemes, it is commonly assumed that the agent is given a feature extractor $\phi : \mathcal{S} \times \mathcal{A} \to \mathbb{R}^d$ which can either be hand-crafted or a pre-trained neural network that transforms a state-action pair to a $d$-dimensional embedding, and the model or the $Q$-function can be predicted by linear functions of the features. In this section, we consider two different kinds of assumptions: a model-based assumption (linear MDP) and a value-based assumption (linear $Q^*$).

**Linear MDP.** The following linear MDP assumption, which was first introduced in [Yang and Wang, 2019, Jin et al., 2019], states that the model of the MDP can be predicted by linear functions of the given features.

**Assumption 2.1** (Linear MDP). *An MDP $\mathcal{M} = (\mathcal{S}, \mathcal{A}, P, r, H, \mu)$ is said to be a linear MDP if the followings hold:*

1. *there are $d$ unknown signed measures $\mu_h = (\mu_h^{(1)}, \mu_h^{(2)}, \ldots, \mu_h^{(d)})$ such that for any $(s, a, s') \in \mathcal{S} \times \mathcal{A} \times \mathcal{S}$, $P_h(s' \mid s, a) = \langle \mu_h(s'), \phi(s, a) \rangle$;*

2. *there exists $H$ unknown vectors $\eta_1, \eta_2, \ldots, \eta_H \in \mathbb{R}^d$ such that for any $(s, a) \in \mathcal{S} \times \mathcal{A}$, $r_h(s, a) = \langle \phi(s, a), \eta_h \rangle$.*

*As in [Jin et al., 2019], we assume for all $(s, a) \in \mathcal{S} \times \mathcal{A}$ and $h \in [H]$, $\|\phi(s, a)\| \leq 1$, $\|\mu_h(S)\|_2 \leq \sqrt{d}$, and $\|\eta\|_2 \leq \sqrt{d}$.*

**Linear $Q^*$.** The following linear $Q^*$ assumption, which is a common assumption in the theoretical RL literature (see e.g. [Du et al., 2019b, 2020]), states that the optimal $Q$-function can be predicted by linear functions of the given features.

**Assumption 2.2** (Linear $Q^*$). *An MDP $\mathcal{M} = (\mathcal{S}, \mathcal{A}, P, r, H, \mu)$ satisfies the linear $Q^*$ assumption if there exist $H$ unknown vectors $\theta_1, \theta_2, \ldots, \theta_H \in \mathbb{R}^d$ such that for any $(s, a) \in \mathcal{S} \times \mathcal{A}$, $Q_h^*(s, a) = \langle \phi(s, a), \theta_h \rangle$. We assume $\|\phi(s, a)\| \leq 1$ and $\|\theta_h\|_2 \leq \sqrt{d}$ for all $(s, a) \in \mathcal{S} \times \mathcal{A}$ and $h \in [H]$.*

---

**Algorithm 1** Reward-Free RL for Linear MDPs: Exploration Phase

---

1: **Input**: Failure probability $\delta > 0$ and target accuracy $\varepsilon > 0$
2: $\beta \leftarrow c_\beta \cdot dH\sqrt{\log(dH\delta^{-1}\varepsilon^{-1})}$ for some $c_\beta > 0$
3: $K \leftarrow c_K \cdot d^3 H^6 \log(dH\delta^{-1}\varepsilon^{-1})/\varepsilon^2$ for some $c_K > 0$
4: **for** $k = 1, 2, \ldots K$ **do**
5:      $Q_{H+1}^k(\cdot, \cdot) \leftarrow 0$ and $V_{H+1}^k(\cdot) = 0$
6:      **for** $h = H, H-1, \ldots, 1$ **do**
7:          $\Lambda_h^k \leftarrow \sum_{\tau=1}^{k-1} \phi(s_h^\tau, a_h^\tau)\phi(s_h^\tau, a_h^\tau)^\top + I$
8:          $u_h^k(\cdot, \cdot) \leftarrow \min\left\{\beta \cdot \sqrt{\phi(\cdot, \cdot)^\top (\Lambda_h^k)^{-1}\phi(\cdot, \cdot)}, H\right\}$
9:          Define the exploration-driven reward function $r_h^k(\cdot, \cdot) \leftarrow u_h^k(\cdot, \cdot)/H$
10:         $w_h^k \leftarrow (\Lambda_h^k)^{-1} \sum_{\tau=1}^{k-1} \phi(s_h^\tau, a_h^\tau) \cdot V_{h+1}^k(s_{h+1}^\tau)$
11:         $Q_h^k(\cdot, \cdot) \leftarrow \min\{(w_h^k)^\top \phi(\cdot, \cdot) + r_h^k(\cdot, \cdot) + u_h^k(\cdot, \cdot), H\}$ and $V_h^k(\cdot) = \max_{a \in \mathcal{A}} Q_h^k(\cdot, a)$
12:         $\pi_h^k(\cdot) \leftarrow \arg\max_{a \in \mathcal{A}} Q_h^k(\cdot, a)$
13:      Receive initial state $s_1^k \sim \mu$
14:      **for** $h = 1, 2, \ldots H$ **do**
15:          Take action $a_h^k \leftarrow \pi^k(s_h^k)$ and observe $s_{h+1}^k \sim P_h(s_h^k, a_h^k)$
16: **return** $\mathcal{D} \leftarrow \{(s_h^k, a_h^k)\}_{(k,h) \in [K] \times [H]}$

---

We note that Assumption 2.2 is weaker than Assumption 2.1. Under Assumption 2.1, it can be shown that for any policy $\pi$, $Q_h^\pi(\cdot, \cdot)$ is a linear function of the given feature extractor $\phi(\cdot, \cdot)$. In this paper, we show that Assumption 2.2 is *strictly weaker* than Assumption 2.1 in the reward-free setting, meaning that reward-free RL under Assumption 2.2 is *exponentially* harder than that under Assumption 2.1.

## 2.3 Reward-Free RL

In the reward-free setting, the goal is to design an algorithm that efficiently explore the state space without the guidance of reward information. Formally, there are two phases in the reward-free setting: *exploration phase* and *planning phase*.

**Exploration Phase.** During the exploration phase, the agent interacts with the environment for $K$ episodes. In the $k$-th episode, the agent chooses a policy $\pi^k$ which induces a trajectory. The agent observes the states and actions $s_1^k, a_1^k, s_2^k, a_2^k, \ldots, s_h^k, a_h^k$ as usual, but does not observe any reward values. After $K$ episodes, the agent collects a dataset of visited state-actions pairs $\mathcal{D} = \{(s_h^k, a_h^k)\}_{(k,h) \in [K] \times [H]}$ which will be used in the planning phase.

**Planning Phase.** During the planning phase, the agent is no longer allowed to interact with the MDP. Instead, the agent is given a set of reward functions $\{r_h\}_{h=1}^H$ where $r_h : \mathcal{S} \times \mathcal{A} \rightarrow [0, 1]$ is the deterministic reward function in level $h$, and the goal here is to output an $\varepsilon$-optimal policy with respect to $r$ using the collected dataset $\mathcal{D}$.

To measure the performance of an algorithm, we define the *sample complexity* to be the number of episodes $K$ required in the exploration phase to output an $\varepsilon$-optimal policy in the planning phase.

## 3 Reward-Free RL for Linear MDPs

In this section, we present our reward-free RL algorithm under the linear MDP assumption.

## 3.1 The Algorithm

The exploration phase of the algorithm is presented in Algorithm 1, and the planning phase is presented in Algorithm 2.

---

**Algorithm 2** Reward-Free RL for Linear MDPs: Planning Phase

---

1: **Input**: Dataset $\mathcal{D} = \{(s_h^k, a_h^k)\}_{(k,h) \in [K] \times [H]}$, reward functions $r = \{r_h\}_{h \in [H]}$
2: $Q_{H+1}(\cdot, \cdot) \leftarrow 0$ and $V_{H+1}(\cdot) = 0$
3: **for** step $h = H, H-1, \ldots, 1$ **do**
4:      $\Lambda_h \leftarrow \sum_{\tau=1}^{K} \phi(s_h^\tau, a_h^\tau) \phi(s_h^\tau, a_h^\tau)^\top + I$
5:      Let $u_h(\cdot, \cdot) \leftarrow \min\left\{ \beta \cdot \sqrt{\phi(\cdot, \cdot)^\top (\Lambda_h)^{-1} \phi(\cdot, \cdot)}, H \right\}$
6:      $w_h \leftarrow (\Lambda_h)^{-1} \sum_{\tau=1}^{K} \phi(s_h^\tau, a_h^\tau) \cdot V_{H+1}(s_{h+1}^\tau, a)$
7:      $Q_h(\cdot, \cdot) \leftarrow \min\{(w_h)^\top \phi(\cdot, \cdot) + r_h(\cdot, \cdot) + u_h(\cdot, \cdot), H\}$ and $V_h(\cdot) = \max_{a \in \mathcal{A}} Q_h(\cdot, a)$
8:      $\pi_h(\cdot) \leftarrow \arg\max_{a \in \mathcal{A}} Q_h(\cdot, a)$
9: **Return** $\pi = \{\pi_h\}_{h \in [H]}$

---

**Exploration Phase.** During the exploration phase of the algorithm, we employ the least-square value iteration (LSVI) framework introduced in [Jin et al., 2019]. In each episode, we first update the parameters $(\Lambda_h, w_h)$ that are used to calculate the $Q$-functions, and then execute the greedy policy with respect to the updated $Q$-function to collect samples. As in [Jin et al., 2019], to encourage exploration, Algorithm 1 adds an upper-confidence bound (UCB) bonus function $u_h$.

The main difference between Algorithm 1 and the one in [Jin et al., 2019] is the definition of the *exploration-driven* reward function. Since the algorithm in [Jin et al., 2019] is designed for the standard RL setting, the agent can obtain reward values by simply interacting with the environment. On the other hand, in the exploration phase of the reward-free setting, the agent does not have any knowledge about the reward function. In our algorithm, in each episode, we design an exploration-driven reward function which is defined to be $r_h(\cdot, \cdot) = u_h(\cdot, \cdot)/H$, where $u_h(\cdot, \cdot)$ is the UCB bonus function defined in Line 8. Note that we divide $u_h(\cdot, \cdot)$ by $H$ so that $r_h(\cdot, \cdot)$ always lies in $[0, 1]$. Intuitively, such a reward function encourages the agent to explore state-action pairs where the amount of uncertainty (quantified by $u_h(\cdot, \cdot)$) is large. After sufficient number of episodes, the uncertainty of all state-action pairs should be low on average, since otherwise the agent would have visited those state-action pairs with large uncertainty as guided by the reward function.

**Planning Phase.** After the exploration phase, the returned dataset contains sufficient amount of information for the planning phase. In the planning phase (Algorithm 2), for each step $h = H, H-1, \ldots, 1$, we optimize a least squares predictor to predict the $Q$-function, and return the greedy policy with respect to the predicted $Q$-function. During the planning phase, we still add an UCB bonus function $u_h(\cdot, \cdot)$ to guarantee optimism and thus correctness of the algorithm (see Lemma 3.3 in the Section 3.2). However, as mentioned above and will be made clear in the analysis, since the agent has acquired sufficient information during the exploration phase, $u_h(\cdot, \cdot)$ should be small on average, which implies the returned policy is near-optimal.

## 3.2 Analysis

In this section we outline the analysis of our algorithm. The formal proof is deferred to the supplementary material. We first give the formal theoretical guarantee of our algorithm.

**Theorem 3.1.** *After collecting $O\left(d^3 H^6 \log(dH\delta^{-1}\varepsilon^{-1})/\varepsilon^2\right)$ trajectories during the exploration phase, with probability $1 - \delta$, our algorithm outputs an $\varepsilon$-optimal policy for an arbitrary number of reward functions satisfying Assumption 2.1 during the planning phase.*

**Remark 3.1.** *If we only have samples of the reward function, we can change Line 6 in Algorithm 2 to $w_h \leftarrow (\Lambda_h)^{-1} \sum_{\tau=1}^{K} \phi(s_h^\tau, a_h^\tau)(V_{H+1}(s_{h+1}^\tau) + r_h^\tau(s_h^\tau, a_h^\tau))$ where $r_h^\tau(s_h^\tau, a_h^\tau)$ is the sampled reward value, and remove $r_h(\cdot, \cdot)$ from Line 7. Our theoretical results still hold after this modification.*

Now we show how to prove Theorem 3.1. Our first lemma shows that the estimated value functions $V^k$ are optimistic with high probability, and the summation of $V_1^k(s_1^k)$ should be small.

**Lemma 3.1.** *With probability $1 - \delta/2$, for all $k \in [K]$,*

$$V_1^*(s_1^k, r^k) \leq V_1^k(s_1^k)$$

*and*

$$\sum_{k=1}^{K} V_1^k(s_1^k) \le c\sqrt{d^3 H^4 K \cdot \log(dKH/\delta)}$$

*for some constant $c > 0$ where $V_1^k(\cdot)$ is as defined in Algorithm 1.*

Note that the definition of the exploration driven reward function $r^k$ used in the $k$-th episode depends only on samples collected during the first $k-1$ episodes. Therefore, the first part of the proof is nearly identical to that of Theorem 3.1 in [Jin et al., 2019]. To prove the second part of the lemma, we first recursively decompose $V_1^k(s_1^k)$ (similar to the standard regret decomposition for optimistic algorithms), and then use the fact that $r_h(\cdot) = u_h(\cdot)/H$ and the elliptical potential lemma in [Abbasi-Yadkori et al., 2012] to given an upper bound on $\sum_{k=1}^{K} V_1^k(s_1^k)$. The formal proof is provided in the supplementary material.

Our second lemma shows that with high probability, if one divides the bonus function $u_h(\cdot, \cdot)$ (defined in Line 5 in Algorithm 2) by $H$ and uses it as a reward function, then the optimal policy has small cumulative reward on average.

**Lemma 3.2.** *With probability $1 - \delta/4$, for the function $u_h(\cdot, \cdot)$ defined in Line 5 in Algorithm 2, we have*

$$\mathbb{E}_{s \sim \mu}\left[V_1^*(s, u_h/H)\right] \le c'\sqrt{d^3 H^4 \cdot \log(dKH/\delta)/K}$$

*for some absolute constant $c' > 0$.*

To prove Lemma 3.2, we first note that $\mathbb{E}_{s \sim \mu}\left[\sum_{k=1}^{K} V_1^*(s, r^k)\right]$ is close to $\sum_{k=1}^{K} V_1^*(s_1^k, r^k)$ by Azuma–Hoeffding inequality and $\sum_{k=1}^{K} V_1^*(s_1^k, r^k)$ can be bounded by using Lemma 3.1. Moreover, for $\Lambda_h$ defined in Line 4 in Algorithm 2, we have $\Lambda_h \succeq \Lambda_h^k$ for all $k \in [K]$ where $\Lambda_h^k$ is defined in Line 7 in Algorithm 1, which implies $u_h(\cdot, \cdot)/H \le r_h^k(\cdot, \cdot)$ for all $k \in [K]$. Therefore, we have

$$\mathbb{E}_{s \sim \mu}\left[V_1^*(s, u_h/H)\right] \le \mathbb{E}_{s \sim \mu}\left[V_1^*(s, r^k)\right]$$

for all $k \in [K]$, which implies the desired result.

Our third lemma states the estimated $Q$-function is always optimistic, and is upper bounded by $r_h(\cdot, \cdot) + \sum_{s'} P_h(s' \mid \cdot, \cdot)V_{h+1}(s')$ plus the UCB bonus function $u_h(\cdot, \cdot)$. The lemma can be proved using the same concentration argument as in [Jin et al., 2019].

**Lemma 3.3.** *With probability $1 - \delta/2$, for an arbitrary number of reward functions satisfying Assumption 2.1 and all $h \in [H]$, we have*

$$Q_h^*(\cdot, \cdot, r) \le Q_h(\cdot, \cdot) \le r_h(\cdot, \cdot) + \sum_{s'} P_h(s' \mid \cdot, \cdot)V_{h+1}(s') + 2u_h(\cdot, \cdot).$$

Now we sketch how to prove Theorem 3.1 by combining Lemma 3.2 and Lemma 3.3. Note that With probability $1 - \delta$, the events defined in Lemma 3.2 and Lemma 3.3 both hold. Conditioning on both events, we have

$$\mathbb{E}_{s \sim \mu}[V_1^*(s, r) - V_1^\pi(s, r)] \le \mathbb{E}_{s \sim \mu}[V_1(s) - V_1^\pi(s, r)]$$
$$\le \mathbb{E}_{s \sim \mu}[V_1^\pi(s, u)] \le \mathbb{E}_{s \sim \mu}[V_1^*(s, u)] \le c'H\sqrt{d^3 H^4 \cdot \log(dKH/\delta)/K},$$

where the first inequality follows by Lemma 3.3, the second inequality follows by Lemma 3.3 and decomposing the $V$-function recursively, the third inequality follows by the definition of $V^*$, and the last inequality follows by Lemma 3.2.

## 4 Lower Bound for Reward-Free RL under Linear $Q^*$ Assumption

In this section we prove lower bound for reward-free RL under the linear $Q^*$ assumption. We show that there exists a class of MDPs which satisfies Assumption 2.2, such that any reward-free RL algorithm requires exponential number of samples during the exploration phase in order to find a near-optimal policy during the planning phase. In particular, we prove the following theorem.

**Theorem 4.1.** *There exists a class of deterministic systems that satisfy Assumption 2.2 with $d = \mathrm{poly}(H)$, such that any reward-free algorithm requires at least $\Omega(2^H)$ samples during the exploration phase in order to find a $0.1$-optimal policy with probability at least $0.9$ during the planning phase for a given set of reward functions $r = \{r_h\}_{h=1}^H$.*

Since deterministic systems are special cases of general MDPs, the hardness result in Theorem 4.1 applies to general MDPs as well. In the remaining part of this section, we describe the construction of the hard instance and outline the proof of Theorem 4.1.

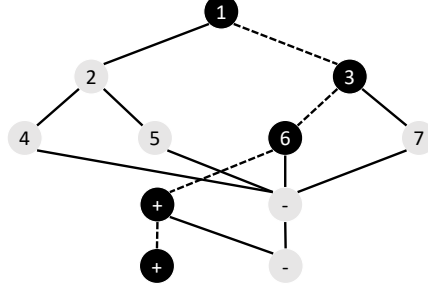

Figure 1: An illustration of the hard instance with $H = 5$. Black states and dashed transitions are those on the optimal trajectory $s_1^*, a_1^*, s_2^*, a_2^*, \ldots, s_{H-1}^*, a_{H-1}^*, s_H^*, a_H^*$.

**State Space and Action Space.**   In the hard instance, there are $H$ levels of states

$$\mathcal{S} = \mathcal{S}_1 \cup \mathcal{S}_2 \cup \ldots \cup \mathcal{S}_H$$

where $\mathcal{S}_h$ contains all states that can be reached in level $h$. The action space $\mathcal{A} = \{0, 1\}$. For each $h \in [H-2]$, we represent each state in $\mathcal{S}_h$ by an integer in $[2^{h-1}, 2^h)$, i.e., $\mathcal{S}_1 = \{1\}$, $\mathcal{S}_2 = \{2, 3\}$, $\mathcal{S}_3 = \{4, 5, 6, 7\}$, etc. We also have $S_{H-1} = \{s_{H-1}^+, s_{H-1}^-\}$ and $S_H = \{s_H^+, s_H^-\}$. The initial states is $1 \in \mathcal{S}_1$.

**Transition.**   For each $h \in [H-3]$, for each $s \in \mathcal{S}_h$, $P_h(s, a)$ is fixed and thus known to the algorithm. In particular, for each $h \in [H-3]$, for each $s \in \mathcal{S}_h$, we define $P_h(s, a) = 2s + a \in \mathcal{S}_{h+1}$ where $a \in \{0, 1\}$. We will define the transition operator for those states $s \in \mathcal{S}_{H-2} \cup \mathcal{S}_{H-1}$ shortly.

**Feature Extractor.**   For each $h \in [H-2]$, for each $(s, a) \in \mathcal{S}_h \times \mathcal{A}$, we define $\phi(s, a) \in \mathbb{R}^d$ so that $\|\phi(s, a)\|_2 = 1$ and for any $(s', a') \in \mathcal{S}_h \times \mathcal{A} \setminus \{(s, a)\}$, we have $|(\phi(s, a))^\top \phi(s', a')| \le 0.01$. In the supplementary material, we use the Johnson–Lindenstrauss Lemma [Johnson and Lindenstrauss, 1984] to show that such feature extractor exists if $d = \mathrm{poly}(H)$. We note that similar hard instance constructions for the feature extractor have previously appeared in [Du et al., 2020]. However, we stress that our construction is different from that in [Du et al., 2020]. In particular, in our hard instance the optimal $Q$-function is exactly linear, while for the hard instance in [Du et al., 2020], the optimal $Q$-function is only approximately linear. Moreover, we focus on the reward-free setting while Du et al. [2020] focused on the standard RL setting.

For all states $s \in \mathcal{S}_{H-1}$, we define

$$\phi(s, a) = \begin{cases} [1, 0, 0, \ldots, 0]^\top & s = s_{H-1}^+, a = 0 \\ [0, 1, 0, \ldots, 0]^\top & s = s_{H-1}^+, a = 1 \\ [0, 0, 0, \ldots, 0]^\top & s = s_{H-1}^- \end{cases}.$$

Finally, for all states $s \in \mathcal{S}_H$, we define

$$\phi(s, a) = \begin{cases} [1, 0, 0, \ldots, 0]^\top & s = s_H^+, a = 0 \\ [0, 0, 0, \ldots, 0]^\top & \text{otherwise} \end{cases}.$$

**The Hard MDPs.** By Yao's minimax principle [Yao, 1977], to prove a lower bound for randomized algorithms, it suffices to define a hard distribution and show that any deterministic algorithm fails for the hard distribution. We now define the hard distribution. We first define the transition operator $P_{H-2}(s, a)$ for those states $s \in \mathcal{S}_{H-2}$. To do this, we first pick a state-action pair $(s^*_{H-2}, a^*_{H-2})$ from $\mathcal{S}_{H-2} \times \mathcal{A}$ uniformly at random, and define

$$P_{H-2}(s, a) = \begin{cases} s^+_{H-1} & s = s^*_{H-2}, a = a^*_{H-2} \\ s^-_{H-1} & \text{otherwise} \end{cases}.$$

To define the transition function $P_{H-1}(s, a)$ for those states $s \in \mathcal{S}_{H-1}$, we pick a random action $a^*_{H-1}$ from $\{0, 1\}$ uniformly at random, and define

$$P_{H-1}(s, a) = \begin{cases} s^+_H & s = s^+_{H-1}, a = a^*_{H-1} \\ s^-_H & \text{otherwise} \end{cases}.$$

**The Reward Functions.** We now define the optimal $Q$-function which automatically implies a set of reward functions $r = \{r_h\}_{h=1}^H$. During the planning phase, the agent will receive $r$ as the reward functions. By construction, there exists a unique trajectory $s^*_1, a^*_1, s^*_2, a^*_2, \ldots, s^*_{H-1}, a^*_{H-1}, s^*_H, a^*_H$ with $(s^*_H, a^*_H) = (s^+_H, 0)$. For each $h \in [H]$, we define $\theta_h$ in Assumption 2.2 as $\phi(s^*_h, a^*_h)/2$. This implies that for each $(s, a) \in \mathcal{S}_H \times \mathcal{A}$,

$$r_H(s, a) = Q^*_H(s, a) = \begin{cases} 0.5 & s = s^*_H, a = a^*_H \\ 0 & \text{otherwise} \end{cases}.$$

For each $(s, a) \in \mathcal{S}_{H-1} \times \mathcal{A}$, we have

$$Q^*_{H-1}(s, a) = \begin{cases} 0.5 & s = s^*_{H-1}, a = a^*_{H-1} \\ 0 & \text{otherwise} \end{cases},$$

which implies that $r_{H-1}(s, a) = 0$ for all $(s, a) \in \mathcal{S}_{H-1} \times \mathcal{A}$. Now for each $h \in [H-2]$, for each $(s, a) \in \mathcal{S}_h \times \mathcal{A}$, we define $r_h(s_h, a_h) = Q^*_h(s_h, a_h) - \max_{a \in \mathcal{A}} Q^*_{h+1}(\cdot, a)$ so that the Bellman equations hold. Moreover, by construction, for each $h \in [H]$, we have $Q^*_h(s, a) = 0.5$ when $(s, a) = (s^*_h, a^*_h)$, and $|Q^*_h(s, a)| \leq 0.01$ when $(s, a) \neq (s^*_h, a^*_h)$ and thus $r_h(\cdot, \cdot) \in [-0.02, 0.5].$[7]

**Proof of Hardness.** Now we sketch the final proof of the hardness result. We define $\mathcal{E}$ to be the event that for all $(s, a) \in \mathcal{D}$ where $\mathcal{D}$ are the state-action pairs collected by the algorithm, we have $s \neq s^*_{H-1} = s^+_{H-1}$. For any deterministic algorithm, we claim that if the algorithm samples at most $2^H/100$ trajectories during the exploration phase, with probability at least 0.9 over the randomness of the distribution of MDPs, $\mathcal{E}$ holds. This is because the feature extractor is fixed and thus the algorithm receives the same feedback before reaching $s^+_{H-1}$. Since there are $2^{H-2}$ state-action pairs $(s, a) \in \mathcal{S}_{H-2} \times \mathcal{A}$ and only one of them satisfies $P_{H-2}(s, a) = s^+_{H-1}$, and the algorithm samples at most $2^H/100$ trajectories during the exploration phase, $\mathcal{E}$ holds with probability at least 0.9.

Now during the planning phase, by construction of the optimal $Q$-function, the only 0.1-optimal policy is $\pi_h(s^*_h) = a^*_h$. However, conditioned on $\mathcal{E}$, any deterministic algorithm correctly output $\pi_{H-1}(s^*_{H-1}) = a^*_{H-1}$ with probability at most 0.5, since conditioned on $\mathcal{E}$, $\mathcal{D}$ does not contain $s^*_{H-1}$, and the set of reward functions $r = \{r_h\}_{h=1}^H$ also does not depend on $a^*_{H-1}$. Therefore, during the planning phase of the algorithm, a 0.1-optimal policy is found with probability at most $0.6 < 0.9$.

## 5 Conclusion

This paper provides both positive and negative results for reward-free RL with linear function approximation. Our results imply three new exponential separations: 1) linear MDP v.s. linear $Q^*$, 2) standard RL v.s. reward-free RL, and 3) query with a simulator v.s. query without a simulator. An interesting future direction is to generalize our results to more general function classes using techniques, e.g., in [Wen and Van Roy, 2013, Ayoub et al., 2020].

## Broader Impact

This work is mainly theoretical. Our theoretical results on reward-free RL with linear function approximation could potentially guide practitioners to design theoretically principled and robust exploration algorithms that can be deployed in practical RL systems.

## Disclosure of Funding

Ruosong Wang and Ruslan Salakhutdinov were supported in part by NSF IIS1763562, US Army W911NF1920104 and ONR Grant N000141812861. Part of the work is done while Simon S. Du was at the Institute for Advanced Study where he was supported by NSF grant DMS-1638352 and the Infosys Membership.

## Footnotes

[5]Du et al. [2019a], Misra et al. [2019] studied the rich-observation setting where the observations are generated from latent states. The latent state dynamics is a tabular one.

[6]We assume that the reward functions are deterministic only for simplicity. Our results can be readily generalized to the case where the rewards are stochastic.

[7] Note that this is slightly different from the assumption that $r_h(\cdot, \cdot) \in [0, 1]$. However, this can be readily fixed by shifting all reward values by 0.02.

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
