[Supplementary Material]

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

# A  Missing Proofs in Section 3

In this section, for all $(k, h) \in [K] \times [H]$, we denote

$$\phi_h^k := \phi(s_h^k, a_h^k).$$

In Algorithm 3 and 2, we recall that

$$\beta = c_\beta dH \sqrt{\log(dH/\delta/\epsilon)}.$$

Since $K = c_K \cdot d^3 H^6 \log(dH\delta^{-1}\varepsilon^{-1})/\varepsilon^2$, we have

$$\beta \geq c_\beta dH \sqrt{\log(dHK/\delta)}$$

for appropriate choices of $c_\beta$ and $c_K$.

## A.1  Proof of Lemma 3.1

To prove Lemma 3.1, we need a concentration lemma similar to Lemma B.3 in [Jin et al., 2019].

**Lemma A.1.** *Suppose Assumption 2.1 holds. Let $\mathcal{E}$ be the event that for all $(k, h) \in [K] \times [H]$,*

$$\left\| \sum_{\tau=1}^{k-1} \phi_h^\tau \left( V_{h+1}^k(s_{h+1}^\tau) - \sum_{s' \in \mathcal{S}} P_h(s'|s_h^\tau, a_h^\tau) V_{h+1}^k(s') \right) \right\|_{(\Lambda_h^k)^{-1}} \leq c \cdot dH \sqrt{\log(dKH/\delta)}$$

*for some absolute constant $c > 0$. Then $\Pr[\mathcal{E}] \geq 1 - \delta/4$.*

*Proof.* The proof is nearly identical to that of Lemma B.3 in [Jin et al., 2019]. The only deference in our case is that we have a different reward functions at different episodes. However, note that in our case

$$r_h^k(\cdot, \cdot) = u_h^k(\cdot, \cdot)/H$$

and hence

$$r_h^k(\cdot, \cdot) + u_h^k(\cdot, \cdot) = (1 + 1/H) \cdot \min \left\{ \beta \cdot \sqrt{\phi(\cdot, \cdot)^\top (\Lambda_h^k)^{-1} \phi(\cdot, \cdot)}, H \right\}.$$

Thus our value function $V_{h+1}^k$ is of the form

$$V(\cdot) := \min \left\{ \max_a w^\top \phi(\cdot, a) + \beta \cdot (1 + 1/H) \cdot \sqrt{\phi(\cdot, a)^\top \Lambda^{-1} \phi(\cdot, a)}, H \right\}$$

for some $\Lambda \in \mathbb{R}^{d \times d}$, and $w \in \mathbb{R}^d$. Therefore, the value function shares exactly the same function class as that in Lemma D.6 in [Jin et al., 2019]. The rest of the proof follow similarly. $\square$

We are now ready to prove Lemma 3.1.

*Proof of Lemma 3.1.* In our proof, we condition on the event $\mathcal{E}$ defined in Lemma A.1, which holds with probability at least $1 - \delta/4$. Since $P_h(s'|s, a) = \phi(s, a)^\top \mu_h(s')$, we have

$$\sum_{s' \in \mathcal{S}} P_h(s'|s, a) V_{h+1}^k(s') = \phi(s, a)^\top \widetilde{w}_h^k$$

where

$$\widetilde{w}_h^k := \sum_{s' \in \mathcal{S}} \mu_h(s') V_{h+1}^k(s')$$

is an unknown vector. By Assumption 2.1, $\sum_{s' \in \mathcal{S}} \mu_h(s') \leq \sqrt{d}$. Therefore,

$$\|\widetilde{w}_h^k\|_2 \leq H\sqrt{d}.$$

We thus have, for all $(h, k) \in [H] \times [K]$ and $(s, a) \in \mathcal{S} \times \mathcal{A}$,

$$\phi(s, a)^\top w_h^k - \sum_{s' \in \mathcal{S}} P_h(s' \mid s, a)^\top V_{h+1}^k(s')$$

$$= \phi(s, a)^\top (\Lambda_h^k)^{-1} \sum_{\tau=1}^{k-1} \phi_h^\tau \cdot V_{h+1}^k(s_{h+1}^\tau) - \sum_{s' \in \mathcal{S}} P_h(s' | s, a) V_{h+1}^k(s')$$

$$= \phi(s, a)^\top (\Lambda_h^k)^{-1} \left( \sum_{\tau=1}^{k-1} \phi_h^\tau V_{h+1}^k(s_{h+1}^\tau) - \Lambda_h^k \widetilde{w}_h^k \right)$$

$$= \phi(s, a)^\top (\Lambda_h^k)^{-1} \left( \sum_{\tau=1}^{k-1} \phi_h^\tau V_{h+1}^k(s_{h+1}^\tau) - \widetilde{w}_h^k - \sum_{\tau=1}^{k-1} \phi_h^\tau (\phi_h^\tau)^\top \widetilde{w}_h^k \right)$$

$$= \phi(s, a)^\top (\Lambda_h^k)^{-1} \left( \sum_{\tau=1}^{k-1} \phi_h^\tau \left( V_{h+1}^k(s_{h+1}^\tau) - \sum_{s'} P_h(s' | s_h^\tau, a_h^\tau) V_{h+1}^k(s') \right) - \widetilde{w}_h^k \right).$$

We have,

$$\left| \phi(s, a)^\top (\Lambda_h^k)^{-1} \left( \sum_{\tau=1}^{k-1} \phi_h^\tau \left( V_{h+1}^k(s_{h+1}^\tau) - \sum_{s'} P_h(s' | s_h^\tau, a_h^\tau) V_{h+1}^k(s') \right) \right) \right|$$

$$= \left| \phi(s, a)^\top (\Lambda_h^k)^{-1/2} (\Lambda_h^k)^{-1/2} \left( \sum_{\tau=1}^{k-1} \phi_h^\tau \left( V_{h+1}^k(s_{h+1}^\tau) - \sum_{s'} P_h(s' | s_h^\tau, a_h^\tau) V_{h+1}^k(s') \right) \right) \right|$$

$$\leq \|\phi(s, a)\|_{(\Lambda_h^k)^{-1}} \cdot \left\| \sum_{\tau=1}^{k-1} \phi_h^\tau \left( V_{h+1}^k(s_{h+1}^\tau) - \sum_{s'} P_h(s' | s_h^\tau, a_h^\tau) V_{h+1}^k(s') \right) \right\|_{(\Lambda_h^k)^{-1}}.$$

By Lemma A.1, we have

$$\left| \phi(s, a)^\top (\Lambda_h^k)^{-1} \left( \sum_{\tau=1}^{k-1} \phi_h^\tau \left( V_{h+1}^k(s_{h+1}^\tau) - \sum_{s'} P_h(s' | s_h^\tau, a_h^\tau) V_{h+1}^k(s') \right) \right) \right|$$

$$\leq cdH \sqrt{\log(dKH/\delta)} \cdot \|\phi(s, a)\|_{(\Lambda_h^k)^{-1}}.$$

Moreover, we have

$$\left| \phi(s, a)^\top (\Lambda_h^k)^{-1} \widetilde{w}_h^k \right| \leq \|\phi(s, a)\|_{(\Lambda_h^k)^{-1}} \cdot \|\widetilde{w}_h^k\|_{(\Lambda_h^k)^{-1}} \leq \|\phi(s, a)\|_{(\Lambda_h^k)^{-1}} \cdot H\sqrt{d}.$$

Therefore, we have

$$\left| \phi(s, a)^\top w_h^k - \sum_{s' \in \mathcal{S}} P_h(s' \mid s, a) V_{h+1}^k(s') \right|$$

$$\leq cdH \sqrt{\log(dKH/\delta)} \cdot \|\phi(s, a)\|_{(\Lambda_h^k)^{-1}} + \|\phi(s, a)\|_{(\Lambda_h^k)^{-1}} \cdot H\sqrt{d}$$

$$\leq c_\beta dH \sqrt{\log(dKH/\delta)} \cdot \|\phi(s, a)\|_{(\Lambda_h^k)^{-1}}$$

$$= \beta \cdot \|\phi(s, a)\|_{(\Lambda_h^k)^{-1}}.$$

Now we prove the first part of the lemma.

**First Part.** Our proof is by induction on $h$. Indeed, for $h = H + 1$, it holds that for all $s \in \mathcal{S}$,

$$V_{H+1}^*(s, r^k) \leq V_{H+1}^k(s)$$

since $V_{H+1}^* = V_{H+1}^k = 0$. Suppose for some $h \in [H]$, it holds that for all $s \in \mathcal{S}$,

$$V_{h+1}^*(s, r^k) \leq V_{h+1}^k(s).$$

Then we have

$$V_h^*(s, r^k) = \max_{a \in \mathcal{A}} \left( r_h^k(s, a) + \sum_{s' \in \mathcal{S}} P_h(s' \mid s, a) V_{h+1}^*(\cdot, r^k) \right)$$

$$\leq \max_{a \in \mathcal{A}} \left( r_h^k(s, a) + \sum_{s' \in \mathcal{S}} P_h(s' \mid s, a) V_{h+1}^k(s', r^k) \right).$$

Notice that for all $(s, a) \in \mathcal{S} \times \mathcal{A}$,

$$\sum_{s' \in \mathcal{S}} P_h(s' \mid s, a)^\top V_{h+1}^k(s', r^k) \leq \phi(s, a)^\top w_h^k + \beta \cdot \|\phi(s, a)\|_{(\Lambda_h^k)^{-1}}.$$

We have

$$V_h^*(s, r^k) \leq \min \left\{ \max_{a \in \mathcal{A}} \left( r_h^k(s, a) + \phi(s, a)^\top w_h^k + \beta \cdot \|\phi(s, a)\|_{(\Lambda_h^k)^{-1}} \right), H \right\} = V_h^k(s)$$

as desired.

**Second Part.** To prove the second part, for all $(k, h) \in [K] \times [H - 1]$, we denote

$$\xi_h^k = \sum_{s' \in \mathcal{S}} P(s'|s_h^k, a_h^k) V_{h+1}^k(s') - V_{h+1}^k(s_{h+1}^k).$$

Conditioned on $\mathcal{E}$,

$$\sum_{k=1}^K V_1^k(s_1^k) \leq \sum_{k=1}^K \left( r_1^k(s_1^k, a_1^k) + \phi(s_1^k, a_1^k)^\top w_h^k + \beta \cdot \|\phi(s_1^k, a_1^k)\|_{(\Lambda_1^k)^{-1}} \right)$$

$$= \sum_{k=1}^K \left( \phi(s_1^k, a_1^k)^\top w_h^k + (1 + 1/H) \cdot \beta \cdot \|\phi(s_1^k, a_1^k)\|_{(\Lambda_1^k)^{-1}} \right)$$

$$\leq \sum_{k=1}^K \left( \sum_{s' \in \mathcal{S}} P(s'|s_1^k, a_1^k) V_2^k(s') + (2 + 1/H) \cdot \beta \cdot \|\phi(s_1^k, a_1^k)\|_{(\Lambda_1^k)^{-1}} \right)$$

$$\leq \sum_{k=1}^K \left( \xi_1^k + V_2^k(s_2^k) + (2 + 1/H) \cdot \beta \cdot \|\phi(s_1^k, a_1^k)\|_{(\Lambda_1^k)^{-1}} \right)$$

$$\leq \dots$$

$$\leq \sum_{k=1}^K \sum_{h=1}^{H-1} \xi_h^k + \sum_{k=1}^K \sum_{h=1}^H (2 + 1/H) \cdot \beta \cdot \|\phi(s_h^k, a_h^k)\|_{(\Lambda_h^k)^{-1}}.$$

Note that for each $h \in [H - 1]$, $\{\xi_h^k\}_{k=1}^K$ is a martingale difference sequence with $|\xi_h^k| \leq H$. Define $\mathcal{E}'$ to be the even that

$$\left| \sum_{k=1}^K \sum_{h=1}^{H-1} \xi_h^k \right| \leq c' H^2 \sqrt{K \log(KH/\delta)}.$$

By Azuma–Hoeffding inequality, we have $\Pr[\mathcal{E}'] \geq 1 - \delta/4$.

Next, we have,

$$\sum_{k=1}^K \sum_{h=1}^H \|\phi(s_h^k, a_h^k)\|_{(\Lambda_h^k)^{-1}} \leq \sqrt{KH \sum_{k=1}^K \sum_{h=1}^H \phi(s_h^k, a_h^k)^\top (\Lambda_h^k)^{-1} \phi(s_h^k, a_h^k)}.$$

By Lemma D.2 in [Jin et al., 2019], we have

$$\sum_{h=1}^H \sum_{k=1}^K \phi(s_h^k, a_h^k)^\top (\Lambda_h^k)^{-1} \phi(s_h^k, a_h^k) \leq 2dH \log(K).$$

Conditioned on $\mathcal{E} \cap \mathcal{E}'$ which holds with probability at least $1 - \delta/2$, we have

$$\sum_{k=1}^{K} V_1^k(s_1^k) \le c'H^2\sqrt{K\log(KH/\delta)} + (2+1/H) \cdot \beta \cdot \sqrt{KH \cdot 2dH\log(K)}$$

$$\le c\sqrt{d^3 H^4 K \cdot \log(dKH/\delta)}$$

for some absolute constant $c > 0$.　　　　　　　　　　　　　　　　$\square$

## A.2　Proof of Lemma 3.2

*Proof of Lemma 3.2.* We denote $\Delta^k = V_1^*(s_1^k, r^k) - \mathbb{E}_{s\sim\mu}[V_1^*(s, r^k)]$. Since $r^k$ depends only on data collected during the first $k-1$ episodes, $\{\Delta^k\}_{k=1}^{K}$ is a martingale difference sequence. Moreover, $|\Delta^k| \le H$ almost surely. Thus, by Azuma-Hoeffding inequality, we have, with probability at least $1 - \delta/8$, there exists an absolute constant $c_1 > 0$, such that

$$\left| \sum_{k=1}^{K} \Delta^k \right| \le c_1 H \sqrt{K\log(1/\delta)},$$

which we condition on in the rest of the proof. Therefore, we have,

$$\mathbb{E}_{s\sim\mu}\left[ \sum_{k=1}^{K} V_1^*(s, r^k) \right] \le \sum_{k=1}^{K} V_1^*(s, r^k) + c_1 H\sqrt{K\log(1/\delta)}.$$

Next, we notice that for all $k \in [K]$,

$$\Lambda_h \succeq \Lambda_h^k.$$

Hence we have for all $(k, h) \in [K] \times [H]$,

$$r_h^k(\cdot, \cdot) \ge u_h(\cdot, \cdot)/H.$$

Hence

$$V_1^*(\cdot, u_h/H) \le V_1^*(\cdot, r_h^k).$$

Together with Lemma 3.1, we have

$$\mathbb{E}_{s\sim\mu}\left[ V_1^*(s, u_h/H) \right] \le \mathbb{E}_{s\sim\mu}\left[ \sum_{k=1}^{K} V_1^*(s, r^k)/K \right] \le K^{-1}\sum_{k=1}^{K} V_1^*(s_1^k, r^k) + c_1 H\sqrt{\log(1/\delta)/K}$$

$$\le c'\sqrt{d^3 H^4 \cdot \log(dKH/\delta)/K}$$

for some absolute constant $c' > 0$.　　　　　　　　　　　　　　　　$\square$

## A.3　Proof of Lemma 3.3

*Proof of Lemma 3.3.* Using the same argument in the proof of Lemma 3.1, with probability at least $1 - \delta/4$, for all $h \in [H]$ and $(s, a) \in \mathcal{S} \times \mathcal{A}$, we have

$$\left| \phi(s, a)^\top w_h - \sum_{s' \in \mathcal{S}} P_h(s' \mid s, a) V_{h+1}(s') \right| \le \beta \cdot \|\phi(s, a)\|_{(\Lambda_h)^{-1}}.$$

Therefore, for all $h \in [H]$ and $(s, a) \in \mathcal{S} \times \mathcal{A}$,

$$Q_h(s, a) \le (w_h)^\top \phi(s, a) + r_h(s, a) + u_h(s, a)$$

$$\le r_h(s, a) + \sum_{s' \in \mathcal{S}} P_h(s' \mid s, a) V_{h+1}(s') + 2\beta \cdot \|\phi(s, a)\|_{(\Lambda_h)^{-1}}.$$

Moreover, $Q_h(s, a) \le H$. Since $u_h(\cdot, \cdot) = \min\left\{ \beta \cdot \sqrt{\phi(\cdot, \cdot)^\top (\Lambda_h)^{-1}\phi(\cdot, \cdot)}, H \right\}$, we have

$$Q_h(s, a) \le r_h(s, a) + \sum_{s'} P_h(s' \mid s, a) V_{h+1}(s') + 2u_h(s, a).$$

Now we prove for all $h \in [H]$ and $(s, a) \in \mathcal{S} \times \mathcal{A}$, $Q_h^*(s, a, r) \leq Q_h(s, a)$. We prove by induction on $h$. When $h = H + 1$ this is clearly true. Suppose for some $h \in [H]$, $Q_{h+1}^*(s, a, r) \leq Q_{h+1}(s, a)$ for all $(s, a) \in \mathcal{S} \times \mathcal{A}$. We have

$$Q_h(s, a) = \min\{(w_h)^\top \phi(s, a) + r_h(s, a) + u_h(s, a), H\}.$$

Since $Q_{h+1}^*(s, a, r) \leq H$ and $u_h(\cdot, \cdot) = \min\left\{\beta \cdot \sqrt{\phi(\cdot, \cdot)^\top (\Lambda_h)^{-1} \phi(\cdot, \cdot)}, H\right\}$, it suffices to prove that

$$Q_{h+1}^*(s, a, r) \leq (w_h)^\top \phi(s, a) + r_h(s, a) + \beta \cdot \|\phi(s, a)\|_{(\Lambda_h)^{-1}}.$$

By the induction hypothesis,

$$\phi(s, a)^\top w_h \geq \sum_{s' \in \mathcal{S}} P_h(s' \mid s, a) V_{h+1}(s') - \beta \cdot \|\phi(s, a)\|_{(\Lambda_h)^{-1}}$$

$$\geq \sum_{s' \in \mathcal{S}} P_h(s' \mid s, a) V_{h+1}^*(s', r) - \beta \cdot \|\phi(s, a)\|_{(\Lambda_h)^{-1}}.$$

Therefore,

$$Q_h^*(s, a, r) = r_h(s, a) + \sum_{s' \in \mathcal{S}} P_h(s' \mid s, a) V_{h+1}^*(s', r)$$

$$\geq (w_h)^\top \phi(s, a) + r_h(s, a) + \beta \cdot \|\phi(s, a)\|_{(\Lambda_h)^{-1}}.$$

$\square$

### A.4 Proof of Theorem 3.1

*Proof of Theorem 3.1.* In our proof we condition on the events defined in Lemma 3.2 and Lemma 3.3 which hold with probability at least $1 - \delta$. By Lemma 3.3, for any $s \in \mathcal{S}$,

$$V_1(s) = \max_{a \in \mathcal{A}} Q_1(s, a) \geq \max_{a \in \mathcal{A}} Q_1^*(s, a, r) = V_1^*(s, r),$$

which implies

$$\mathbb{E}_{s_1 \sim \mu}[V_1^*(s_1, r) - V_1^\pi(s_1, r)] \leq \mathbb{E}_{s_1 \sim \mu}[V_1(s_1) - V_1^\pi(s_1, r)].$$

Note that

$$\mathbb{E}_{s_1 \sim \mu}[V_1(s_1) - V_1^\pi(s_1, r)]$$
$$= \mathbb{E}_{s_1 \sim \mu}[Q(s_1, \pi_1(s_1)) - Q_1^\pi(s_1, \pi_1(s_1), r)]$$
$$= \mathbb{E}_{s_1 \sim \mu, s_2 \sim P_1(\cdot|s_1, \pi_1(s_1))}[r_1(s_1, \pi_1(s_1)) + V_2(s_2) + u_1(s_1, \pi(s_1)) - r_1(s_1, \pi_1(s_1)) - V_2^\pi(s_2)]$$
$$= \mathbb{E}_{s_1 \sim \mu, s_2 \sim P_1(\cdot|s_1, \pi_1(s_1))}[V_2(s_2) + u_1(s_1, \pi(s_1)) - V_2^\pi(s_2)]$$
$$= \mathbb{E}_{s_1 \sim \mu, s_2 \sim P_1(\cdot|s_1, \pi_1(s_1)), s_3 \sim P_2(\cdot, |s_2, \pi_2(s_2))}[u_1(s_1, \pi(s_1)) + u_2(s_2, \pi(s_2)) + V_3(s_3) - V_3^\pi(s_3)]$$
$$= \ldots$$
$$= \mathbb{E}_{s \sim \mu}[V_1^\pi(s, u)].$$

By definition of $V_1^*(s, u)$, we have

$$\mathbb{E}_{s \sim \mu}[V_1^\pi(s, u)] \leq \mathbb{E}_{s \sim \mu}[V_1^*(s, u)].$$

By Lemma 3.2,

$$\mathbb{E}_{s \sim \mu}[V_1^*(s, u)] = H \cdot \mathbb{E}_{s \sim \mu}[V_1^*(s, u/H)] \leq c' H \sqrt{d^3 H^4 \cdot \log(dKH/\delta)/K}.$$

By taking $K = c_K \cdot d^3 H^6 \log(dH\delta^{-1}\varepsilon^{-1})/\varepsilon^2$ for a sufficiently large constant $c_K > 0$, we have

$$\mathbb{E}_{s_1 \sim \mu}[V_1^*(s_1, r) - V_1^\pi(s_1, r)] \leq H \cdot \mathbb{E}_{s \sim \mu}[V_1^*(s, u/H)] \leq c' H \sqrt{d^3 H^4 \cdot \log(dKH/\delta)/K} \leq \varepsilon,$$

which implies $\pi$ is $\varepsilon$-optimal with respect to $r$. $\square$

---

**Algorithm 3** Reward-Free RL under Linear $Q^*$: Exploration Phase

---
1: **for** $h = 1, 2, \ldots, H$ **do**
2:     Find $(s_h^1, a_h^1), (s_h^2, s_h^2), \ldots, (s_h^d, a_h^d)$ such that $\phi(s_h^1, a_h^1), \phi(s_h^2, s_h^2), \ldots, \phi(s_h^d, a_h^d)$ form a set
    of linear basis of span $\left(\{\phi(s, a)\}_{(s,a) \in \mathcal{S} \times \mathcal{A}}\right)$
3:     **for** $i = 1, 2, \ldots, d$ **do**
4:         Query $t_h^i \leftarrow P_h(s_h^i, a_h^i)$
5: **return** $\mathcal{D} \leftarrow \{(s_h^i, a_h^i, t_h^i)\}_{(i,h) \in [d] \times [H]}$

---

---

**Algorithm 4** Reward-Free RL under Linear $Q^*$: Planning Phase

---
1: **Input**: Dataset $\mathcal{D} = \{(s_h^i, a_h^i, t_h^i)\}_{(i,h) \in [d] \times [H]}$, reward functions $r = \{r_h\}_{h \in [H]}$
2: $Q_{H+1}(\cdot, \cdot) \leftarrow 0$ and $V_{H+1}(\cdot) = 0$
3: **for** step $h = H, H-1, \ldots, 1$ **do**
4:     **for** $i = 1, 2, \ldots, d$ **do**
5:         $Q_h(s_h^i, a_h^i) \leftarrow r_h(s_h^i, a_h^i) + V_{h+1}(t_h^i)$
6:     $Q_h(s, a) \leftarrow \sum_{i=1}^d \beta_i \cdot Q_h(s_h^i, a_h^i)$ if $\phi(s, a) = \sum_{i=1}^d \beta_i \cdot \phi(s_h^i, a_h^i)$
7:     $V_h(\cdot) = \max_{a \in \mathcal{A}} Q_h(\cdot, a)$
8:     $\pi_h(\cdot) \leftarrow \arg\max_{a \in \mathcal{A}} Q_h(\cdot, a)$
9: **Return** $\pi = \{\pi_h\}_{h \in [H]}$

---

## B   Reward-Free RL under Linear $Q^*$ Assumption with a Simulator

In this section, we present an algorithm for reward-free RL under the linear $Q^*$ assumption (Assumption 2.2) in deterministic systems, when the agent has access to a generative model (a.k.a. simulator) of the MDP. More specifically, for each state action $(s, a) \in \mathcal{S} \times \mathcal{A}$, for each $h \in [H]$, we assume the agent can query $P_h(s, a)$. We show that after querying the transition operator for polynomial number of times during the exploration phase, during the planning phase, the agent can find an optimal policy for any given reward function $r$.

The exploration phase of our algorithm is described in Algorithm 3, while the planning phase is described in Algorithm 4.

During the exploration phase, for each level $h$, we find $(s_h^1, a_h^1), (s_h^2, s_h^2), \ldots, (s_h^d, a_h^d)$ such that $\phi(s_h^1, a_h^1), \phi(s_h^2, s_h^2), \ldots, \phi(s_h^d, a_h^d)$ form a set of linear basis of span $\left(\{\phi(s, a)\}_{(s,a) \in \mathcal{S} \times \mathcal{A}}\right)$ by querying the feature extractor $\phi$. Then we query $P_h(s_h^i, a_h^i)$ for each $i \in [d]$. During the planning phase, for each $(i, h) \in [d] \times [H]$, we calculate $Q_h(s_h^i, a_h^i) = r_h(s_h^i, a_h^i) + V_{h+1}(P_h(s_h^i, a_h^i))$ by the Bellman equation. For each $(s, a) \in \mathcal{S} \times \mathcal{A}$, we can always find $\beta$ such that $\phi(s, a) = \sum_{i=1}^d \beta_i \cdot \phi(s_h^i, a_h^i)$, since $\phi(s_h^1, a_h^1), \phi(s_h^2, s_h^2), \ldots, \phi(s_h^d, a_h^d)$ form a set of linear basis of span $\left(\{\phi(s, a)\}_{(s,a) \in \mathcal{S} \times \mathcal{A}}\right)$. Due to the linearity of the optimal $Q$-function, we set $Q_h(s, a) = \sum_{i=1}^d \beta_i \cdot Q_h(s_h^i, a_h^i)$. We define the $V$-function and the policy accordingly.

Notice that during the exploration phase, the algorithm query the transition operator for $dH$ times in total. To prove the correctness, we prove by induction on $h$ that during the planning phase, $Q_h(\cdot, \cdot) = Q_h^*(\cdot, \cdot)$. Note that this is clearly true when $h = H + 1$. Suppose $Q_{h+1}(\cdot, \cdot) = Q_{h+1}^*(\cdot, \cdot)$. It is clear that $V_{h+1}(\cdot) = V_{h+1}^*(\cdot)$, which implies $Q_h(s_h^i, a_h^i) = Q_h^*(s_h^i, a_h^i)$ by the Bellman equation. By Assumption 2.2, if $\phi(s, a) = \sum_{i=1}^d \beta_i \cdot \phi(s_h^i, a_h^i)$,

$$Q_h(s, a) = \sum_{i=1}^d \beta_i \cdot Q_h(s_h^i, a_h^i) = \sum_{i=1}^d \beta_i \cdot Q_h^*(s_h^i, a_h^i) = Q_h^*(s, a).$$

## C   Missing Proofs in Section 4

In this hard instance construction in Section 4, for each $h \in [H - 2]$, for each $(s, a) \in \mathcal{S}_h \times \mathcal{A}$, we define $\phi(s, a) \in \mathbb{R}^d$ so that $\|\phi(s, a)\|_2 = 1$ and for any $(s', a') \in \mathcal{S}_h \times \mathcal{A} \setminus \{(s, a)\}$, we have

$|(\phi(s, a))^{\top} \phi(s', a')| \leq 0.01$. The following lemma demonstrates the existence of such feature extractor.

**Lemma C.1.** *There exists a set of vectors $\{\phi_1, \phi_2, \ldots, \phi_{2^H}\} \subset \mathbb{R}^d$ with $d = \text{poly}(H)$ such that*

1. $\|\phi_i\| = 1$ *for all $i \in [2^H]$;*

2. $|\phi_i^{\top} \phi_j| \leq 0.01$ *for all $i, j \in [2^H]$ with $i \neq j$.*

*Proof.* This is a direct implication of Lemma A.1 in [Du et al., 2020] by setting $n = 2^H$ and $\varepsilon = 0.01$. $\qquad\square$

Note that the above lemma implies the existence of the required feature exactor, since for each $h \in [H - 2]$, there are less than $2^H$ state-action pairs in $\mathcal{S}_h \times \mathcal{A}$. We simply define the feature of the $i$-th state-action pair in $\mathcal{S}_h \times \mathcal{A}$ to be $\phi_i$ in the above lemma.

*Proof of Theorem 4.1.* In order to prove Theorem 4.1, by Yao's minimax principle [Yao, 1977], it suffices to prove that for the hard distribution constructed in Section 4, for any deterministic algorithm $\mathcal{A}$ that samples at most $2^H/100$ trajectories during the exploration phase, the probability (over the randomness of the hard distribution) that $\mathcal{A}$ outputs a $0.1$-optimal policy in the planning phase is at most $0.9$.

We first show that for the deterministic algorithm $\mathcal{A}$, among all the $2^{H-2}$ choices for $(s_{H-2}^*, a_{H-2}^*)$, $s_{H-1}^+$ is in the collected dataset $\mathcal{D}$ for at most $2^H/100$ choices for $(s_{H-2}^*, a_{H-2}^*)$ during the exploration phase. Note that whenever $(s_{H-2}, a_{H-2}) \neq (s_{H-2}^*, a_{H-2}^*)$, we must have $s_{H-1} = s_{H-1}^-$ and $s_H = s_H^-$. Therefore, the feedback received by $\mathcal{A}$ is always the same unless $(s_{H-2}, a_{H-2}) = (s_{H-2}^*, a_{H-2}^*)$. However, since $\mathcal{A}$ samples at most $2^H/100$ trajectories during the exploration phase, there are most $2^H/100$ choices for $(s_{H-2}^*, a_{H-2}^*)$ during the exploration phase for which $s_{H-1}^+$ is in the collected dataset $\mathcal{D}$.

Recall that $\mathcal{A}$ is deterministic. For any choice of $(s_{H-2}^*, a_{H-2}^*)$, if $s_{H-1}^+$ is not in the collected dataset $\mathcal{D}$, the collected dataset $\mathcal{D}$ is always the same, no matter $a_{H-1}^* = 0$ or $a_{H-1}^* = 1$. Moreover, for any fixed choice of $(s_{H-2}^*, a_{H-2}^*)$, it can be verified that the reward function $r$ does not depend on the choice of $a_{H-1}^*$. Note that during the planning phase, algorithm $\mathcal{A}$ deterministically maps the collected dataset $\mathcal{D}$ and the reward function $r$ to a policy. Furthermore, the only $0.1$-optimal policy must satisfy $\pi(s_h^*) = a_h^*$. However, for any choice of $(s_{H-2}^*, a_{H-2}^*)$, if $s_{H-1}^+$ is not in the collected dataset $\mathcal{D}$, $\pi(s_{H-1}^*)$ does not depend on $a_{H-1}^*$ since both the collected dataset $\mathcal{D}$ and the reward function $r$ do not depend on $a_{H-1}^*$. Therefore, for those choices of $(s_{H-2}^*, a_{H-2}^*)$, $\mathcal{A}$ outputs a $0.1$-optimal policy with probability at most $0.5$. Therefore, the probability that $\mathcal{A}$ outputs a $0.1$-optimal policy is at most

$$\frac{2^H/100}{2^{H-2}} + \left(1 - \frac{2^H/100}{2^{H-2}}\right)/2 \leq 0.6.$$

$\square$