[Reviews · NeurIPS 2020]

Review 1

Summary and Contributions: The authors in this paper study the problem of learning in reward free fashion, to construct a good estimation of the model that can be further used for planing for a class of possible reward. This problem has been studied in tabular settings in prior works, and this work extends them to the value-based setting where the transitions and rewards are linear in some feature representation, resulting in Q function to be linear for any policy. The authors focus on fixed horizon MDPs The contribution of this paper is mainly two-fold, 1) In the linear model case, they show that an optimism based approach gives rise to a sample efficient algorithm. In another word, the algorithm runs in a polynomial in dimension and 1/epsilon number of episodes, and then, given any reward function satisfying linear MDP, it outputs epsilon optimal policy for that reward. 2) The second contribution of the paper is on the lower bound. The authors show that, if the Q function of just optimal policy is linear, the sample complexity would be exponential in dimensions.

Strengths: This work's strength is in its theoretical development and proposing an algorithm that learns an explicit model useful for later planning. Also, the lower bound. The authors well motivate the problem, and the approach is sound.

Weaknesses: Maybe not a limitation, but the results in this paper are mainly a direct extension of Jin 2020, which is on tabular MDP and Jin 2019, which is one of the later studies on MDPs with linear Q functions. It makes the novelty of the techniques and results in this paper limited, but it should not stop the paper from being accepted. It is absolutely alright to have a paper like this, which states a series of strong results but directly built on the top of prior works. The biggest concern of mine about this paper is the lack of rigor. The authors did not define S, what is A, what are possible admittable Q function, what rewards are allowed, and what transitions are permitted. It is quite disappointing that authors did not specify non of these spaces, and whether Q even exists, let alone existence of an optimal policy or that the "max" even applies here.

Correctness: Since the analyses are directly borrowed from prior works, I believe the author did a great job in transferring the results to their study, and also the advancement they made are sound. I was not fully convinced in the lower bound part. I am not fully convinced that if the Q^* is linear no only on the optimal action, but also on the suboptimal action, then the lower bound is exponential. I can see it when Q^* is linear for a^*, but not for suboptimal one. So I could not verify that part. I did not check the appendix for that but would have been great if the authors provide more details on the lower bound in the main text.

Clarity: The paper is clear and I enjoyed reading it.

Relation to Prior Work: The authors did a good job of referring to prior works. Maybe the authors find it useful to discuss also this paper: Active Model Estimation in Markov Decision Processes, it seems relevant.

Reproducibility: Yes

Additional Feedback:


Review 2

Summary and Contributions: The authors develop algorithms and theoretical analysis for reward-free RL in a setting with linear MDP's and linear function approximation extending previous work on reward-free RL which was in the tabular setting. They provide an algorithm for RL which has polynomial trajectory sample complexity in the dimension of the feature space and the horizon of the episode. Interestingly, there is no dependence on the size of the action or state space. They also prove a lower bound result by explicitly showing a class of deterministic MDPs such that the sample complexity for any RL algorithm has to be exponential.

Strengths: There are a number of significant achievements in this work. First, their algorithm goes beyond the tabular setting. Second it gives a sample complexity that is independent of the number of states and actions and polynomial in the other parameters. This is significant because a polynomial dependence in the number of states is fatal and that is the best one can do in the tabular case. A very interesting result is the lower bound result which they are able to exhibit even with just deterministic MDPs. The proofs are rigourous and the formulation of the framework and the background is very good.

Weaknesses: The restriction to the linear MDP setting is a weakness. I think that the extension to more general results is not too far off, but one cannot tell unless one tries. The proofs rely strongly on this restriction.

Correctness: Yes, this is a very competent theoretical analysis. There was no experimental backing.

Clarity: Yes.

Relation to Prior Work: Yes.

Reproducibility: Yes

Additional Feedback:


Review 3

Summary and Contributions: The paper extends the reward-free approach recently introduced by Jin et al '20 to RL exploration in low-rank / linear MDPs, and provides lower bounds as well. However, in terms of techniques I feel the paper does not overall make a solid leap forward, but rather extends prior ideas. ======= Unchanged after rebuttal

Strengths: - The extension to the linear MDP setting of reward free exploration. - The lower bound (which I find more interesting) shows the separation between generative and online RL with this objective and with respect to the single reward-function.

Weaknesses: I find the extension of reward-free exploration to the linear MDP setting to be fairly straightforward, reusing most of the available techniques e.g. in Jin '20 for exploration on linear MDPs. Although reward free exploration has been only recently proposed, in general there are ways to create reward-free versions of the algorithms fairly easily (as it happens in this paper). While there has been a growing number of papers focussing on linear MDPs, it is the reviewer's opinion that we should not ``overfit'' to this setting by replicating / extending all the available bandit / tabular results here. The lower bound again seems to be a small variations of the ideas of Du '20 (but this is acknowledged by the authors)

Correctness: The paper looks correct to my understanding

Clarity: yes

Relation to Prior Work: yes

Reproducibility: Yes

Additional Feedback: I would suggest to put more emphasis on the lower bound as opposed to the upper bound. The upper bound looks fairly straightforward to obtain. Why do you need optimism in the planning phase?


Review 4

Summary and Contributions: The paper extends recent work in reward-free RL (where an agent must explore its environment without reward signal, and then later try planning to solve a task with a given reward function) to the linear function approximation case. The paper proposes an algorithm for solving this problem that is provably efficient under certain assumptions. From my understanding, the algorithm essentially takes different ideas from existing work [Jin et al, 2019, 2020] and combines them. In particular, it is shown that for linear MDPs, the sample complexity of the resulting algorithm is polynomial, whereas with less restrictive assumptions it is exponential. This has implications for the “hardness” of reward-free RL under certain assumptions (i.e. that linear optimal value functions are a strictly weaker assumption than linear MDPs), as well as potentially the difference between normal RL and reward-free RL.

Strengths: The paper combines two existing pieces of work to develop an efficient algorithm to tackle the reward-free RL problem. Extended this beyond the tabular case is important, since now we can start thinking beyond small gridworlds and tackle continuous state MDPs. I thought the paper was very well written, and the insights from the theory were well presented, and would probably be of interest to the community. The fact that the assumption about the linearity of the optimal value function is weaker than the linearity of the MDP itself is unsurprising but good to know.

Weaknesses: I realise that the paper and contribution here are 100% theoretical in nature. However, it would have been nice to see even a synthetic toy empirical experiment which could showcase the algorithms in practice. Perhaps an illustration of Theorem 3.1, where delta, epsilon, H are varied and the corresponding number of samples increases?

Correctness: The theory all appears sound to me, but I was unable to verify all of the maths.

Clarity: The paper was generally well written. There were some minor typos and missing words, but nothing too major. I particularly liked the contributions and insights section at the end of the introduction. I also liked the sketch explanations of the proofs for the various lemmas and theorems. I found page 8 very hard to read since it was so dense, but I’m not sure if there’s any solution to that. It may be helpful If it could be explained roughly the intuition behind the “hard” transition distributions. I also found the use of the word “levels” a bit confusing. I understand it in the context of line 263, but is it not more easily understood as just timestep (particularly in Section 2.1). This would just be worth clarifying.

Relation to Prior Work: I am not well-versed in this exact field, but the related work seems comprehensive. The authors may be interested in some *very* recent work which slightly modifies the problem and reduces Jin et al’s dependency from O(S^2) to O(S) [1]. You could also include [2] on line 96. [1] Task-agnostic Exploration in Reinforcement Learning. Zhang et al, arxiv [2] Chentanez, Nuttapong, Andrew G. Barto, and Satinder P. Singh. "Intrinsically motivated reinforcement learning." Advances in neural information processing systems. 2005.

Reproducibility: Yes

Additional Feedback: 1. I would just like to confirm my understanding of the algorithmic contributions of this work. As far as I understand, Jin et al [2019] propose a learning algorithm for the standard RL case with linear function approximation in linear MDPs. Then Jin et al [2020] propose a method for efficient exploration in the reward-free RL case. This is for normal MDPs but in the tabular setting. In that work, exploration is achieved by constructing a reward function where the reward is 1 for states that are “significant”, and 0 otherwise, and then solving the resulting task with an efficient learning algorithm. In this paper, the idea is to use the learning algorithm from Jin et al [2019]. However, that was in the normal RL case with rewards, and here there are no rewards. So a reward function is constructed similar to Jin et al [2020], but instead of 1/0, it’s a UCB based score that captures uncertainty. And then finally that “task” is solved, which is easier because of the linearity in the MDP. 2. Looking beyond this, in the planning phase, the agent is given the explicit reward function. In practice, however, it’s probably the case that the agent just gets samples from the reward function. In that setting, what would the impact on the framework be? Presumably it would still be pretty efficient because of the efficient exploration, and it would just require observing samples to estimate the underlying reward function? It would also be interesting to find out what happens when the linearity assumption is assumed to hold, but doesn’t (or only approximately holds), but I certainly don’t think the paper needs to deal with that question. 3. On line 182-183, this is the number of episodes for *any* reward function (e.g. the most adversarial one), correct? 4. What is the effect of increasing/decreasing c_\Beta? 5. Minor typos: a. 22: algorithm -> algorithms b. 37: deal WITH both… c. 55: both A polynomial… d. 115: use citep e. 290: reward functionS ========== POST-REBUTTAL ========== Thanks to the authors for their response. Although the work is somewhat incremental with respect to Jin et al [2020], I still think it makes good progress on the "reward free RL" problem. It would have been nice to see at least a small scale empirical result, but I think because this is so theoretically-focused, it's not necessary. The linearity assumption is quite strong, so it would be useful to have some thoughts in the final version as to how we can go about loosening this assumption going forward.

[Author Response · NeurIPS 2020]

We thank all the reviewers for their valuable feedback and appreciating our contributions. First we would like to
emphasize the technical novelty of our upper bound and lower bound as Reviewer #1, Reviewer # 3 and Reviewer #4
commented on the technical novelty of our theoretical results.

**Technical novelty of the upper bound.** In the exploration phase, Jin et al. [2020] set reward to be 1 for significant
states and 0 for other states. Note their technique cannot be used in Linear MDPs because there are possibly infinitely
many states and thus one needs to take the structure of linear functions into account. In this paper, we use UCB bonus
*as the reward signal* in the exploration phase. To our knowledge, this idea is new in the literature. We also would like to
thank Reviewer # 4 for a detailed description of our main algorithmic ideas.

**Technical novelty of the lower bound.** We have discussed the differences between our lower bound and that in [Du
et al. 2020] in Line 276 - 279. We acknowledge that in our hard instance, we use a similar feature extractor as that in
[Du et al. 2020]. However, all other aspects of the hard instance construction are significantly different from that in [Du
et al. 2020]. For example, for the hard instance in [Du et al. 2020], only a single state-action pair has non-zero reward
value, which is not case in our hard instance. Note that such distinction is crucial, since in our hard instance the optimal
$Q$-function is *exactly* linear, whereas the the optimal $Q$-function is only *approximately* linear in the hard instance in [Du
et al. 2020]. Moreover, we focus on the reward-free setting while Du et al. [2020] focused on the standard RL setting.

Below we address specific concerns from each reviewer.

—— **To Reviewer #1** ——

**Lack of rigor.** We have introduced necessary background on MDP in Section 2.1, including the state space, the action
space, the transition operator, the reward distribution, the $Q$-function, etc. We have also provided necessary definitions
related to linear function approximation in Section 2.2. Our descriptions mostly follow existing works. We will expand
this part to make the paper clearer.

$Q^*$ **is linear on the suboptimal action.** In our construction, when defining the reward functions, we first define the
optimal $Q$-function ($Q^*$) as a specific linear function (see Line 292), and then define the reward values according to
the Bellman equations (see Line 296). Therefore, the optimal $Q$ function must be linear for both optimal actions and
suboptimal actions in our hard instances.

**Relation to prior work.** We will discuss the suggested paper in the next version. Thanks for the suggestion.

—— **To Reviewer #2** ——

**Extension to more general settings.** Even in the standard RL setting, going beyond linear MDPs is hard. See the open
problems in [Du et al. 2020]. Therefore, we believe it is highly non-trivial to obtain more general results.

—— **To Reviewer #3** ——

**More emphasize on the lower bound.** Thanks for the suggestion. We will emphasize more on the lower bound and
the implied conceptual messages in the final version.

**Why do you need optimism in the planning phase.** Optimism in the planning phase is used when we prove Lemma
3.3. It also guarantees the correctness of the first inequality in Line 247-248.

—— **To Reviewer #4** ——

We would like to thank the reviewer for the detailed description of our key ideas in our algorithm. The understanding is
correct.

**Experiments.** Thanks for the suggestion. We will consider adding empirical results in the next version.

**Related work.** Thanks for the references. We will add more discussion in the next version.

**Agent just gets samples from the reward function.** If we only have samples, we can change Line 6 in Algorithm 2 to
$w_h \leftarrow (\Lambda_h)^{-1} \sum_{\tau=1}^K \phi(s_h^\tau, a_h^\tau)(V_{h+1}(s_{h+1}^\tau) + r_h^\tau(s_h^\tau, a_h^\tau))$ where $r_h^\tau(s_h^\tau, a_h^\tau)$ is the *sampled* reward value, and remove
$r_h(\cdot, \cdot)$ from Line 7. Our theoretical results still hold after this modification, and we will add a discussion on this.

**Linearity approximately holds.** This is an interesting question and we will list it as a future direction.

**Line 182-183.** This is correct.

**The effect of increasing/decreasing $c_\beta$.** $c_\beta$ needs to be larger than a universal constant in order to guarantee optimism.
Once $c_\beta$ is larger than that constant, the sample complexity decreases as $c_\beta$ decreases.

[Meta-Review · NeurIPS 2020]

The authors study sequential decision processes without reward function. The goal is to learn the transition dynamics such that various reward functions could be optimised efficiently in the future. The authors extend recent work to the linear function approximation case. They provide an analysis of the sample complexity, and show that while for linear MDPs complexity is polynomial, this is not true for MDPs with a linear optimal value functions, providing insight on the hardness of this second class of problems. The strengths of the paper are the theoretical development of the algorithm and the lower bound for MDPs with linear optimal Q functions. The proofs are rigorous. A limitation of the paper is that it is relatively close to the earlier work on reward-free RL in the tabular case and MDPs with linear Q functions. This limits the novelty. One reviewer suggested adding a (toy) experiment to better understand the practical implications of the theory. Although all reviewers agreed that the paper is somewhat incremental, most reviewers agreed that the results are nevertheless strong and of interest to the community. Especially the lower bound was an interesting insight to many of the reviewers. Thus, I’d like to recommend this paper for acceptance. In their rebuttal, the authors have clarified several aspects of the paper. I’d like to ask the authors to add these clarifications to the final draft of the paper.